# AN EMPIRICAL STUDY OF THE EFFECT OF BACKGROUND DATA SIZE ON THE STABILITY OF SHAPLEY ADDITIVE EXPLANATIONS (SHAP) FOR DEEP LEARNING MODELS

**Han Yuan**[*] **& Mingxuan Liu**[*] **& Lican Kang**
Duke-NUS Medical School
National University of Singapore
`{yuan.han, mingxuan.liu}@u.duke.nus.edu`
`kanglican@duke-nus.edu.sg`

**Chenkui Miao**
The First School of Clinical Medicine
Nanjing Medical University
`miaochenkui@njmu.edu.cn`

**Ying Wu**
School of Statistics and Data Science
Nankai University
`ywu@nankai.edu.cn`

## ABSTRACT

SHapley Additive exPlanations (SHAP) is a popular method that requires a background dataset in uncovering the deduction mechanism of artificial neural networks (ANNs). Generally, a background dataset consists of instances randomly sampled from the training dataset. However, the sampling size and its effect on SHAP remain unexplored. In this work, we empirically explored the effect and illustrated several tips when applying SHAP. The code is publicly accessible[1].

## 1 INTRODUCTION & METHOD

Nowadays, interpreting model inference is becoming as important as techniques for enhancing model accuracy (Lauritsen et al., 2020). Some machine learning (ML) models like the decision tree can be easily understood by humans while others like ANNs are too complex to be interpreted even by experts (Lundberg & Lee, 2017; Xie et al., 2022). To address this problem, researchers proposed various explanations: instance-level and model-level. The instance-level explanation typically provides feature contribution analyses for a single prediction, and the model-level explanation provides feature importance across all predictions.

SHAP provides both instance and model-level explanations through SHAP values and variable rankings (Lundberg & Lee, 2017). SHAP values are the direct production from SHAP calculations while variable rankings are measured by the sum of each variable's absolute SHAP values across all instances (Lundberg et al., 2020). To eliminate computing complexity, the SHAP package[2] includes various explainers for different ML models. DeepExplainer is an efficient explainer for ANNs and requires a background dataset to serve as a prior expectation for the instances to be explained.

While the official SHAP documentation[3] suggests 100 randomly drawn samples from the training data as an adequate background dataset, other studies employed different sampling sizes (van der Velden et al., 2020; Giese et al., 2021; Kaufmann et al., 2021). This conflict raises questions: What is the recommended background data size on SHAP explanations and is there an impact of using different data sizes? In a pilot experiment, we found that both instance-level SHAP values and model-level variable rankings fluctuate when we adopt random sampling to obtain background datasets with small

---

[*]These authors contributed equally.
[1]`https://github.com/Han-Yuan-Med/shap-bg-size`
[2]`https://pypi.org/project/shap/`
[3]`https://shap.readthedocs.io/en/latest/`

sizes. To quantify such fluctuation and better answer the questions above, we conducted an empirical study with different background dataset sizes applied to interpret three-layer ANNs (See Appendix A). We then quantified the effect on instance-level explanations (SHAP values). Lastly, we used an exact (BLEU score (Papineni et al., 2002)) and fuzzy (Jaccard index (Fletcher & Islam, 2018)) approach for evaluating model-level explanations (variable rankings).

## 2 EXPERIMENTS & CONCLUSION

The fluctuation of SHAP local and global explanations originates from changes in SHAP values, as assessed by a statistical variance measure. We observe that the variance sum per variable across instances in the explanation set decreases as the background sample size increases (See Appendix Table 5). We further utilized the quartile-based BLEU and Jaccard index to compute the exact and fuzzy model-level stabilities. Tables 1 and 2 demonstrate that the pairwise analyses of the different rankings resulted in improved BLEU and Jaccard scores when the background dataset size increased. Additionally, the BLEU and Jaccard index values showed a U-shape stability across Quartiles 1 to 4.

Table 1: Quartile-based BLEU results

| m | Average | Quartile 1 | Quartile 2 | Quartile 3 | Quartile 4 |
|---|---------|------------|------------|------------|------------|
| 100 | 0.432 | 0.478 | 0.269 | 0.360 | 0.619 |
| 500 | 0.557 | 0.594 | 0.387 | 0.509 | 0.739 |
| 1000 | 0.644 | 0.657 | 0.476 | 0.624 | 0.818 |

m: the sample size of the background dataset

Table 2: Quartile-based Jaccard index results

| m | Average | Quartile 1 | Quartile 2 | Quartile 3 | Quartile 4 |
|---|---------|------------|------------|------------|------------|
| 100 | 0.868 | 0.903 | 0.787 | 0.832 | 0.950 |
| 500 | 0.904 | 0.934 | 0.833 | 0.876 | 0.975 |
| 1000 | 0.924 | 0.936 | 0.855 | 0.911 | 0.993 |

Our empirical study quantifies the stability of SHAP explanations at both the instance and model-level and complements its robustness exploration (Lakkaraju et al., 2020). We present a positive relationship between background dataset sample size and the stability of SHAP explanations: More coherent SHAP values and variable rankings were observed when larger background datasets were used. This phenomenon could be partially explained by inference of the central limit theorem: The background dataset converges to the overall distribution at the standard rate of the root of sample size (Rosenblatt, 1956). Therefore, sampling with a larger size could lead to less randomness, a more representative background dataset, and more stable SHAP explanations (Li & Ding, 2017). Furthermore, our results suggested that the optimal background dataset size depends on a user's expectation with regard to the exactness of the ranking. For example, in our pairwise analysis using the BLEU score, we did not observe unchanged variable order rankings, even at large background data sizes. Therefore, the concrete variable importance ranking by SHAP requires careful consideration. On the other hand, we saw stable rankings using a fuzzy comparison (Jaccard index), even at small background dataset sizes. Particularly, the U-shape of the comparative BLEU and Jaccard scores indicates that SHAP is more reliable in ranking the most and least important variables compared to the moderately important ones.

A large background dataset (even the whole training dataset) leads to coherent SHAP explanations. However, larger background datasets lead to higher computation budgets. To estimate the upper limit of an affordable background sample size, users are recommended to conduct a pilot experiment using a small background dataset (e.g., size of 100) to estimate the computational complexity. Given a complexity $C_{100}$ derived from a background dataset with 100 samples, the complexity $C_m$ using a background dataset with $m$ samples can be approximated by $\frac{m}{100} \times C_{100}$ based on the linear relationship between background sample size and computational complexity (Lundberg & Lee, 2017). After the determination of the background sample size, users are suggested to select a representative cohort by sampling from high-density areas or using K-means clustering (Kim et al., 2016).

URM STATEMENT

We acknowledge that Han Yuan, Mingxuan Liu, and Chenkui Miao meet the URM criteria.

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

## A   METHOD DETAILS

### A.1   SHAP VALUES AND VARIABLE RANKINGS

SHAP provides instance-level and model-level explanations by SHAP value and variable ranking. In a binary classification task (the label is 0 or 1), the inputs of an ANN model are variables $var_{i,j}$ from an instance $D_i$, and the output is the prediction probability $P_i$ of $D_i$ of being classified as label 1. In general, we are interested in interpreting a stack of instances $D$ at both the instance and model levels.

$$D = \{D_i\}, i = 1, \ldots, N \tag{1}$$

$$D_i = \{var_{i,j}\}, j = 1, \ldots, V \tag{2}$$

$$P_i = ANN(D_i), i = 1, \ldots, N \tag{3}$$

where $i$ represents the $i$-th instance in $D$ and $j$ stands for the $j$-th variable in $D_i$.

DeepExplainer, the function for SHAP calculations in an ANN, provides instance-level explanations of $P_i$ through the contribution ($shap_{i,j,bg}$) of each variable ($var_{i,j}$) to the prediction deviation from a prior $P_{bg}$, which is the probability expectation of samples in the background dataset $bg$.

$$P_i - P_{bg} = \sum_{j=1}^{V} shap_{i,j,bg} \tag{4}$$

With the instance-level $shap_{i,j,bg}$, we then compute the model-level variable importance $I_{j,bg}$ (Lundberg & Lee, 2017) as follows:

$$I_{j,bg} = \sum_{i=1}^{N} |shap_{i,j,bg}| \tag{5}$$

where absolute SHAP values for a particular variable are summed over all instances in $D$. Based on variable importance, users obtain the variable rankings effortlessly (larger $I_{j,bg}$, higher importance and ranking).

As shown in formula 4 and 5, a background dataset is a prerequisite for DeepExplainer and its application on ANNs interpretation. Generally, a background dataset consists of instances randomly sampled from the training data (Lundberg & Lee, 2017). However, the use of small background sample sizes potentially causes fluctuation of SHAP explanations, which may affect users' trust in the one-shot SHAP explanations.

While the statistical variance is well suited for evaluating the fluctuations of SHAP values, we need more refined measures for assessing changes in variable rankings. Specifically, as changes in variable rankings can be seen as equivalent to word order changes, we adopt the BLEU assessment, typically used to compare the word order of a translated text with regard to a reference translation, for measuring differences in variable rankings. Another potential evaluation comes from similarity computation in set theory: the Jaccard index.

### A.2   FLUCTUATION QUANTIFICATION OF VARIABLE RANKING

#### A.2.1   BLEU FOR EXACT FLUCTUATION EVALUATIONS

BLEU (Papineni et al., 2002) aims for evaluating machine translation by comparing n-gram matches between the candidate translation and the reference translation. An n-gram is a contiguous sequence of n items from a given text. The first step in BLEU is to compute the n-gram matches precision score $p_n$: the matched n-gram counts in the candidate translation over the number of candidate n-grams in the reference translation. The second step is to multiply $p_n^{\frac{1}{2^n}}$ ($1 \leq n \leq k$, $n$ is an integer) with penalty terms: $\exp(\min(0, 1 - \frac{l_c}{l_r}))$, where $k$ is the maximum number of words in the matched subsequence, and $l_c$ and $l_r$ stand for lengths of the candidate translation and the reference translation, respectively.

$$BLEU = \exp\left(\min\left(0, 1 - \frac{l_c}{l_r}\right)\right) \prod_{n=1}^{k} p_n^{\frac{1}{2^n}} \tag{6}$$

As stated above, if we treat one variable ranking as "the candidate translation" and the other as "the reference translation", we can also compute the difference between these two rankings. The formula 6 is modified to suit our application: (1) Since the ranking sequences are of equal length and contain the same elements, the $\exp(\min(0, 1 - \frac{l_c}{l_r}))$ and $p_1$ are always 1; (2) Only two-grams BLEU is used to quantify the relevant sequence between two rankings; (3) Considering that two-grams BLEU might fail to detect the drastic order change of two-grams units, we proposed quartile-based computation where the variable ranking is split into four quartiles, $p_{2,q}^{\frac{1}{4}}$ is calculated for each quartile $q$ ($q = 1, 2, 3, 4$), and the average value $\text{BLEU}_Q$ serves as the final assessment value.

$$\text{BLEU}_Q = \frac{1}{4} \times \sum_{q=1}^{4} p_{2,q}^{\frac{1}{4}} \tag{7}$$

We use a fictive example to clarify $\text{BLEU}_Q$ (See Table 3). There are two rankings: Ranking #1 [a, b, c, d, e, f, g, h, i, j, k, l] and Ranking #2 [a, c, b, d, e, f, h, g, i, j, k, l]. We first split the two rankings into four parts. For example, Ranking #1 is split into [a, b, c], [d, e, f], [g, h, i], and [j, k, l]. Then we calculate two-grams in each quartile of these two rankings. Ranking #1's first part is [a, b, c] and the corresponding two-grams are [a, b] and [b, c]. With two-grams in each quartile, we can easily obtain the matched two-grams counts and the number of two-grams. Finally, the $\text{BLEU}_Q$ here is $\frac{1}{4} \times (0^{\frac{1}{4}} + 1^{\frac{1}{4}} + 0^{\frac{1}{4}} + 1^{\frac{1}{4}}) = 0.25$ according to formula 7.

Table 3: Two-grams and the corresponding precision score of Ranking #1 and #2

| Quartile | Ranking #1 | Ranking #2 | Matched Num | Reference Num | Precision Score |
|---|---|---|---|---|---|
| 0-25% | [a, b], [b, c] | [a, c], [c, b] | 0 | 2 | 0/2 = 0 |
| 25-50% | [d, e], [e, f] | [d, e], [e, f] | 2 | 2 | 2/2 = 1 |
| 50-75% | [g, h], [h, i] | [h, g], [g, i] | 0 | 2 | 0/2 = 0 |
| 75-100% | [j, k], [k, l] | [j, k], [k, l] | 2 | 2 | 2/2 = 1 |

Matched Num: the number of matched two-gram units between two variable rankings
Reference Num: the number of two-gram units in one variable ranking

### A.2.2 JACCARD INDEX FOR FUZZY FLUCTUATION EVALUATIONS

In contrast to $\text{BLEU}_Q$ which focuses on the exact match of two-gram units, Jaccard index (Fletcher & Islam, 2018), defined as the size of the intersection divided by the size of the union of the sample sets, is used for evaluating the fuzzy similarity between sample sets. Given that any ranking in our study contains the same variables, the Jaccard index cannot be directly used.

Like $\text{BLEU}_Q$, we propose the quartile-based $\text{Jaccard}_Q$, wherein the variable ranking is split into four quartiles, $Jaccard_q$ is calculated in each quartile $q$ ($q = 1, 2, 3, 4$), and the average value across all quartiles works as a final assessment of the fluctuation of variable rankings.

$$\text{Jaccard}_Q = \frac{1}{4} \times \sum_{q=1}^{4} Jaccard_q \tag{8}$$

The same fictive sample is used for clarification (See Table 4): We first split the two rankings into four parts, then compute the intersection, union number, and Jaccard index in each quartile, and finally obtain the mean value of all Jaccard indexes $\frac{1}{4} \times (1 + 1 + 1 + 1) = 1$.

Table 4: Variable subsets and corresponding Jaccard index of Ranking #1 and #2

| Quartile | Ranking #1 | Ranking #2 | Intersection Num | Union Num | Jaccard index |
|---|---|---|---|---|---|
| 0-25% | a, b, c | a, c, b | 3 | 3 | 3/3 = 1 |
| 25-50% | d, e, f | d, e, f | 3 | 3 | 3/3 = 1 |
| 50-75% | g, h, i | h, g, i | 3 | 3 | 3/3 = 1 |
| 75-100% | j, k, l | j, k, l | 3 | 3 | 3/3 = 1 |

Intersection Num: the number of variables in the intersection between these two rankings
Union Num: the number of variables in the union between the two rankings

## A.3 DATASET AND MODEL ARCHITECTURE

We implemented an empirical study of SHAP stability using a de-identified intensive care unit dataset. This dataset includes 44,918 admission episodes (including 3,958 positive episodes, defined as admissions within patient mortality) of the Beth Israel Deaconess Medical Center (Johnson et al., 2016). We randomly separated the data set into development and explanation sets. The development set consisted of 31,442 (70%) patients, and the explanation set was made up of 13,476 (30%) patients. The development set was used to develop the ANN and to generate background datasets. The explanation set was put aside to be interpreted by SHAP. The variables to be ranked included heart rate, age, respiration rate, systolic blood pressure, diastolic blood pressure, mean arterial pressure, white blood cell count, platelet count, glucose, sodium, lactate, bicarbonate, blood urea nitrogen, creatinine, chloride. An ANN with three layers was used as a backbone model in this study because no substantial gain was observed with more layers. The ANN was made up of 2 hidden layers with 128 and 64 rectified linear units respectively and 1 output layer using sigmoid activation.

## A.4 EMPIRICAL STUDY SETTING

We varied background data size from 100 to 1,000 and performed 100 simulations under each background data size. In each simulation, a background dataset with a fixed size was sampled from the training dataset. Then SHAP values and variable rankings are calculated on the explanation set. After 100 simulations, we obtained 100 SHAP values for each variable in a single instance and applied statistical variance to depict the fluctuation of SHAP values in this instance: For variable $var_j$, its variance sum is $\sum_{i=1}^{N} \frac{1}{99} \sum_{bg=1}^{100} \left( |shap_{i,j,bg}| - \frac{1}{100} \sum_{bg=1}^{100} |shap_{i,j,bg}| \right)^2$. Also, we received $p = 100$ variable rankings and $C_p^2$ different pairs of rankings in the model level. $\mathrm{BLEU_Q}^k$ and $\mathrm{Jaccard_Q}^k$ represents the quartile-based $\mathrm{BLEU_Q}$ and $\mathrm{Jaccard_Q}$ index of the $k$-th pair, respectively. Then the mean of $\mathrm{BLEU_Q}^k$ and $\mathrm{Jaccard_Q}^k$ across all pairs were calculated to assess the fluctuation of variable rankings. All computations were carried out using PyTorch version 1.6.0, Python version 3.8, and R version 4.0.3.

$$\text{Mean BLEU} = \sum_{k=1}^{C_p^2} \left( \mathrm{BLEU_Q^k} \right); \text{Mean Jaccard} = \sum_{k=1}^{C_p^2} \left( \mathrm{Jaccard_Q^k} \right) \tag{9}$$

## B ADDITIONAL EXPERIMENTS

Table 5: The mean variance of SHAP values across all observations

| Variables | Variance sum (m=100) | Variance sum (m=500) | Variance sum (m=1000) |
|---|---|---|---|
| Age | 153.31 | 22.16 | 11.21 |
| Heart rate | 26.71 | 8.85 | 5.20 |
| Systolic blood pressure | 271.31 | 44.27 | 24.13 |
| Diastolic blood pressure | 74.64 | 11.22 | 5.52 |
| Arterial pressure | 169.92 | 22.91 | 13.32 |
| Respiration rate | 33.08 | 5.75 | 2.85 |
| Glucose | 72.18 | 16.35 | 7.32 |
| Bicarbonate | 10.17 | 2.02 | 1.11 |
| Creatinine | 56.95 | 7.92 | 3.82 |
| Chloride | 67.52 | 7.85 | 3.17 |
| Lactate | 22.63 | 6.93 | 3.05 |
| Platelet | 325.33 | 49.46 | 23.21 |
| Blood urea nitrogen | 212.63 | 30.46 | 15.60 |
| Sodium | 60.20 | 6.54 | 2.65 |
| White blood cells | 83.88 | 10.40 | 5.71 |

m: the sample size of the background dataset

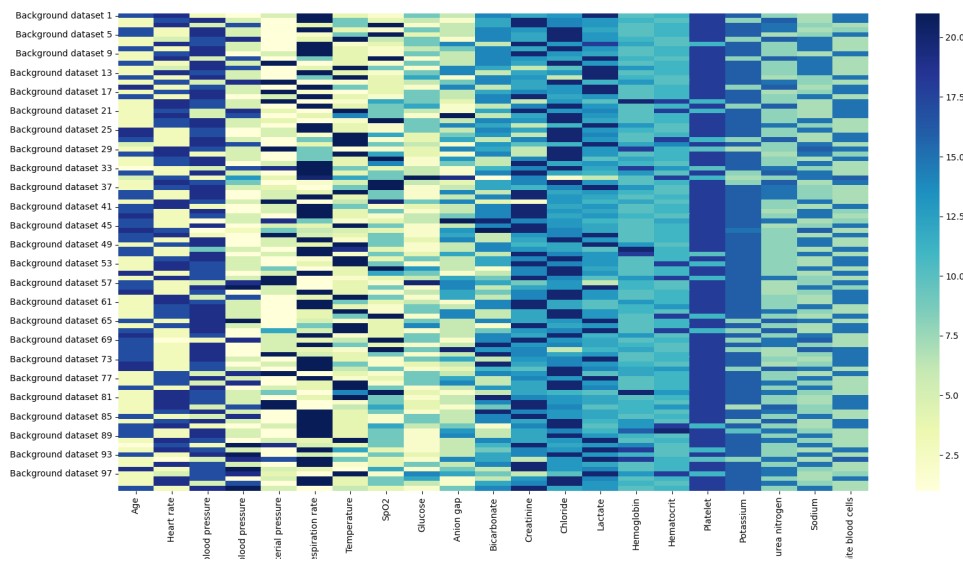

(a) The variable rankings using 100 background dataset samples.

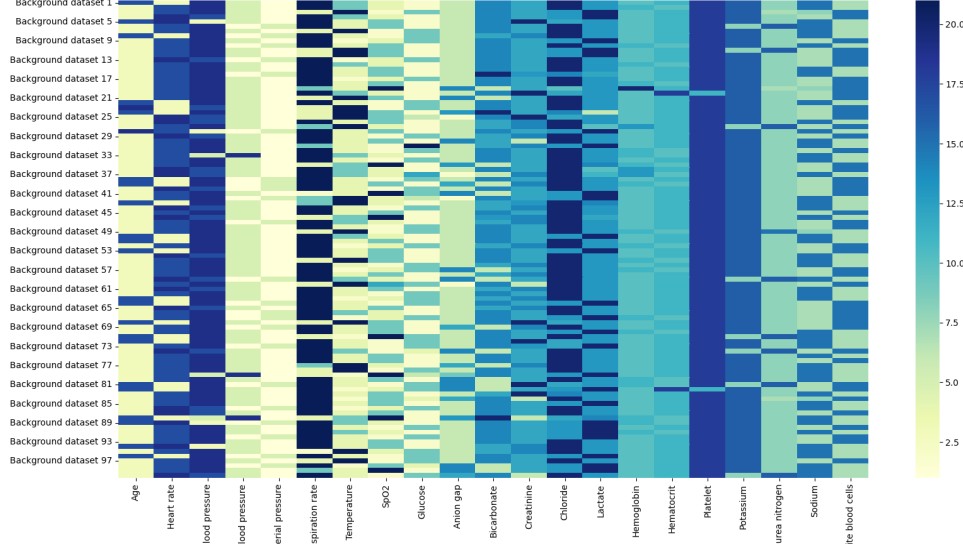

(b) The variable rankings using 1,000 background dataset samples.

Figure 1: Fluctuation of variable rankings using background size of 100 and 1,000. Based on 100 simulations, we obtained 100 variable rankings for each of the two background sizes. Each row corresponds to one simulation; each column represents a variable's ranking order across the 100 simulations. The color bar on the right indicates the variable ranking sequence (blue means low rankings and yellow stands for high rankings). The small color blocks in each sub-figure (b) column change less than those of (a), indicating that smaller volatilities appear with larger background sizes.

