# OpenReview forum: "An Empirical Study of the Effect of Background Data Size on the Stability of SHapley Additive exPlanations (SHAP) for Deep Learning Models"
_ICLR.cc/2023/TinyPapers — Submitted to Tiny Papers @ ICLR 2023_

### Official Review · Reviewer_U9JN · 2023-03-22

**Confidence:** 3

**Summary Of Contributions:**

The article considers the impact of background dataset size for computing Shapley values in the SHapley Additive exPlanations (SHAP) method. The authors provide preliminary experiments to understand this impact along with some pointers for selecting the background data size.

**Rating:**

Clear, Correct, and Reproducible (CCR): a submission which meets the reviewing criteria

**Strengths And Weaknesses:**

The article considers the impact of background data size while using SHapley Additive exPlanations (SHAP) for explaining the impact of input variables in Neural Networks.

Strengths:
- As expected, higher background dataset sizes tend to be stable by virtue of lower variability in their estimates.
- The authors also evaluate the impact of background dataset size on SHAP by comparing two predicted variable impact rankings using the BLEU index and Jaccard index that can compare same-length sequences. Generally, both measures provide similar assessments that increasing background size is good.
- Interestingly, fuzzy measures where an exact match of rankings is not important tend to perform better than exact rankings. This indicates some limitations of SHAP and provides a good way to measure SHAP or similar methods.

Weaknesses:
- The paper demonstrates some insights into SHAP. However, there are few unique actionable insights (besides the pilot study) for leveraging or improving SHAP. Furthermore, there does not seem to be a way to use SHAP for smaller datasets. Further details in the suggested changes.
- Minor concern: Are the values in tables 1 and 2 for the two indices supposed to be the same?

**Suggested Changes:**

- Extending the study to study simple methods that can leverage SHAP for smaller background datasets can be useful. Can fuzzily similar variable rankings be leveraged to get a better estimate of the true variable rankings?
- Experiments can be extended to larger and more complex networks. Furthermore, new experiments should focus on the small background dataset scenario.
- Comparing SHAP rankings with other explainability rankings (not based on SHAP) will be useful for seeing how well SHAP performs.
- Finally, new insights (without pilot studies) into the impact of background size will significantly improve this article.

---

> ### Author Response · Authors · 2023-04-18
> **Tips from the empirical study**
>
> Thank you for the detailed comments and questions. We have fixed the table duplication typo in the updated version. Our empirical study quantifies the stability of SHAP explanations at both the instance (SHAP values) and model-level (variable rankings) and points to a positive relationship between background sample size and the stability of SHAP explanations. More coherent SHAP values and variable rankings were observed when larger background datasets were used. To estimate the upper limit of an affordable background sample size, users are recommended to conduct a pilot experiment using a small background dataset (e.g., size of 100) to estimate the computational complexity. Also, our results suggested that the optimal background dataset size depends on a user’s expectation of the ranking exactness. In our pairwise analysis using the BLEU score, we did not observe exact replicate variable order rankings while stable rankings using a fuzzy ranking comparison (Jaccard index) were noted even at small background sizes. Therefore, while SHAP is a trustworthy method for evaluating variable importance, the concrete variable ranking requires careful consideration. Additionally, the U-shape of the comparative BLEU and Jaccard scores indicates that SHAP is more reliable in ranking the most and least important than the moderately important variables.

---

### Official Review · Reviewer_CzF3 · 2023-03-28

**Confidence:** 2

**Summary Of Contributions:**

This paper studies the impact of the number of samples on the behavior of SHapley Additive exPlanations (SHAP), an approach to interpret predictive models. The paper studies the role of the number of samples used to estimate SHAP and provides guidance on how to estimate the number of samples needed to use SHAP in a meaningful manner

**Rating:**

Great Start (GS): a submission which meets some of the reviewing criteria but has room for improvement

**Strengths And Weaknesses:**

**Strengths**

- The paper studies the number of samples needed in a systematic manner. This is a good problem to solve especially as there is no consensus on what constitutes a correct setup
- The initial experiment with number of samples and per variable variance provides evidence that there is a problem related to the correct use of SHAP measure. A deeper study can lead to recommendations that can be very useful to the ML community, specifically for researchers/practitioners working on model interpretability
- Code provided to support reproducibility

**Weaknesses**

- The reviewer is a novice with SHAP values. So I was not able to see a clear connection between the number of samples experiment and the BLEU/Jaccard scores shown in Tables 1 and 2. I hope the authors would consider making the connection between the main idea and these clear to help a broader audience appreciate their work

**Suggested Changes:**

- I observed that the values in Table 1 and Table 2 are identical. Is this correct or expected?
- The paper is off to a great start. Adding connections between results in main text (Table 1 & Table 2) and SHAP values seen in experiments would make the paper accessible to a broader audience.

---

> ### Author Response · Authors · 2023-04-18
> **BLEU score and Jaccard index for assessing the stability of SHAP explanations**
>
> Thank you for the feedback. We have fixed the table duplication typo in the updated version. For evaluating the fluctuations of SHAP values, the statistical variance is well-suited. Variable rankings are another explanation based on SHAP values. For assessing changes in them, we need more refined measures. Specifically, as changes in variable rankings can be seen as equivalent to word order changes, we adopt the BLEU assessment, typically used to compare the word order of a translated text with regard to a reference translation, for measuring differences in variable rankings in Table 1. Another potential evaluation comes from similarity computation in set theory: the Jaccard index in Table 2.

---

### Comment · Area_Chair_FcPz · 2023-06-04
**Revised version**

This work meets the threshold for archival, contains the URM statement and is deanonymized

---

### Meta-Review · Area_Chair_FcPz · 2023-04-08

**Recommendation:** Invite to revise
**Confidence:** 4

**Metareview:**

The impact of the number of samples on the behaviour of SHAP and the role of the number of samples used to estimate SHAP is considered. The authors provide preliminary experiments to understand this impact and discuss sample size selection.

Vital topic, where there are still a lot of gaps in the literature. The initial experiment with the number of samples and per variable variance provides evidence of a problem related to correctly using the SHAP measure. Code provided to support reproducibility. Tables 1 and 2 are the same values? Is this a typo, or is there a reason behind it?

The paper is off to a great start. The main issue is the lack of connection with the main text and results. Given the page limit, there is no need for more experiments, but please try and expand the discussions. Get the msg across. Due to the results in Tables 1 and 2 being the same and without explanation, there is no clear indication if this is an error. This paper can improve if the authors can explain and take the reviewers' suggestions.


**Summary:**

The impact of the number of samples on the behaviour of SHAP and the role of the number of samples used to estimate SHAP is considered. The authors provide preliminary experiments to understand this impact and discuss sample size selection.

**Comments And Feedback To The Authors:**

Please read the reviewer's comments and consider revising and resubmitting.

**Reason For Not Giving A Higher Recommendation:**

There is room for improvement. And Authors do need to check their tables to make sure they are correct.

**Reason For Not Giving A Lower Recommendation:**

N/A

---

> ### Author Response · Authors · 2023-04-18
> **Take-home message from the paper**
>
> Thank you for the detailed comments and questions. We have fixed the table duplication typo in the updated version. The main take-home message in this paper is that: SHAP is a popular and useful tool and users should take care when applying it to deep learning models, especially in the selection of background data. In this paper, we try to answer two questions: What is the recommended background data size on SHAP explanations and is there an impact of using different data sizes? Through an empirical study, we have shown that SHAP explanations fluctuate when using a small background dataset and that these fluctuations decrease when the background dataset sampling size increases. This finding holds true for both instance and model-level explanations.

---

### Decision · Program_Chairs · 2023-04-09

**Decision:**

Invite to archive

**Comment:**

Please correct the Table 1 / 2 duplication mistake.

---

> ### Author Response · Authors · 2023-04-18
> **Correct duplication mistake**
>
> Thank you for the feedback. The mistake has been corrected in the updated version.